# ENHANCING ROBUSTNESS OF VISUAL OBJECT LOCALIZATION BY INTRODUCING RETINA-INSPIRED MAPPING TO CONVOLUTIONAL NEURAL NETWORKS

## ABSTRACT

Foveated vision, a trait shared by many animals including humans, has yet to be fully exploited in machine learning applications despite its key contributions to biological visual function. In this study, we investigate whether retinotopic mapping, a critical component of foveated vision, can improve image categorization and localization performance when incorporated into deep convolutional neural networks (CNNs). In particular, we incorporated retinotopic mapping into the inputs of classic off-the-shelf CNNs and retrained these network on the IMAGENET task. Surprisingly, the retinotopically mapped network performed equally well in classification but showed improved robustness to arbitrary image zooms and rotations, especially for isolated objects. In addition, this network showed improved classification localization when the foveated center of the transform was moved, mimicking a key capability of the human visual system that is lacking in standard CNNs. These results suggest that retinotopic mapping may underlie important preattentive visual processes.

## 1 ROBUSTNESS OF CONVOLUTIONAL NEURAL NETWORKS

Deep learning algorithms have made tremendous progress in the last few years. For some visual recognition tasks, such as the IMAGENET challenge (Russakovsky et al., 2015), they have achieved superhuman accuracy. However, one limitation that remains is their vulnerability to adversarial attacks. Studies have shown that these learned models can be fooled by imperceptible modifications to images that are undetectable to humans (Huang et al., 2017). Yet these small distortions cause the algorithms to incorrectly classify examples with high confidence (Szegedy et al., 2013). This vulnerability makes deep networks unstable and unsuitable for use in safety-critical domains like medicine, autonomous vehicles, or other life-or-death situations. Before deep learning can be relied upon for such applications, researchers must find ways to make these models more resilient to adversarial examples and introduce human-level robustness to ensure that mistakes do not have dangerous consequences in the real world.

### 1.1 ATTACKING CNNs WITH A GEOMETRICAL ROTATION

While previous adversarial attacks have focused on imperceptible pixel perturbations, such approaches introduce changes in many independent dimensions without reflecting the perturbations that occur in natural environments. We present a more ecologically valid attack: controlled image rotations. Just as real-world objects appear in different orientations, we evaluate the robustness of models to rotations of test inputs. Compared to modifying independent pixels, rotation represents a coherent, whole-image transformation controlled by a single parameter, the rotation angle. Our goal is to investigate attacks that simulate realistic rotations in order to understand vulnerabilities and inform robust algorithm development.

To evaluate such a rotation-based attack, we examined popular off-the-shelf CNNs pre-trained on standard, large image datasets. For each test image, we first computed the unperturbed baseline accuracy. We then systematically rotated the images and tracked the change in model loss. To perform such an attack on a model $m$, we follow this simple procedure. Given an image $I$ and the output of the model $\boldsymbol{p} = m(I)$, which returns a probability vector over $K = 1000$ classes, the loss

function $\mathcal{L}$ is defined as the cross-entropy between the predicted probability vector and the ground truth label $y$: $\mathcal{L}(m(I), y)$. This is the loss that was minimized during the training procedure using gradient descent. By denoting a rotation of the image by an angle $\theta$ as $\mathrm{rot}(I; \theta)$, we define the rotation-based attack as the following heuristic:

$$\bar{\theta} = \arg\max_{\theta} \mathcal{L}(m(\mathrm{rot}(I; \theta)), y) \tag{1}$$

$$\hat{y} = \arg\max_{k}(\bar{p}_k) \text{ with } \bar{\boldsymbol{p}} = m(\mathrm{rot}(I; \bar{\theta})) \tag{2}$$

so that we can compute the concordance between the predicted label $\hat{y}$ for the image rotated at the angle $\bar{\theta}$ with the worst loss with respect to the ground truth label $y$. Using this procedure, we calculated the overall accuracy on the entire test set, quantifying the network's brittleness to natural image rotations.

Our experiments showed that while VGG16 and RESNET101 achieved 55% and 74% baseline accuracy, respectively, rotation attacks significantly reduced this. The application of the maximally deceptive rotation to each image reduced the accuracy to 4% and 39%, respectively. These results were confirmed by performing the same procedure on the ANIMAL 10K (Yu et al., 2021) dataset (see Appendix Figure 5). In addition, we computed for any given rotation the average accuracy over the test dataset, showing that this average accuracy dropped sharply compared to baseline accuracy, confirming that this simple geometrical transformation mislead the networks. The decline was steepest between two cardinal rotations, demonstrating limited rotational invariance compared to humans (Rousselet et al., 2003).

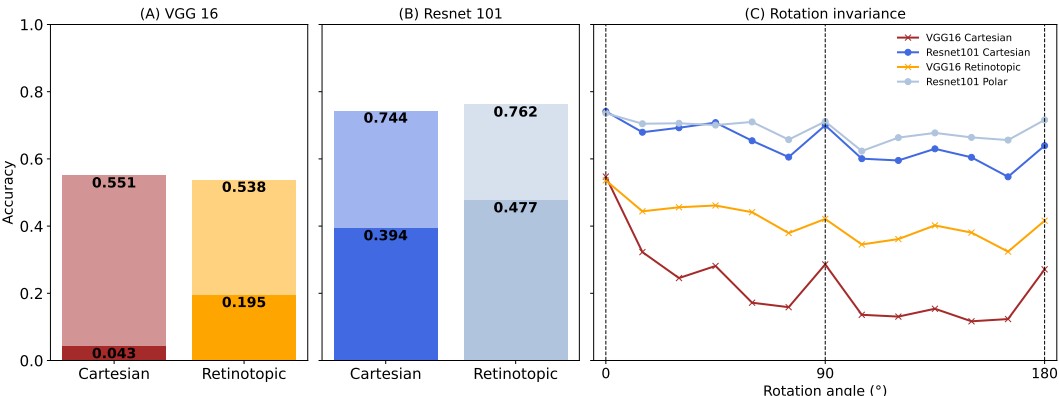

Figure 1: To illustrate the brittleness of CNNS, we show the effect of rotation-based attack by plotting accuracy over a sample of 1000 images from the IMAGENET (Russakovsky et al., 2015) dataset, for **(A)** VGG16 and **(B)** RESNET101 networks. For all the networks, we plot the accuracy over the raw data set (without rotation, in light color), or on the image rotated at the angle $\bar{\theta}$ with the worst loss (rotation-based attack, in full color). **(C)** Average accuracy over a sample of 1500 images from the IMAGENET (Russakovsky et al., 2015) dataset, shown for both retrained and pre-trained networks with different input image rotations. The rotation is applied around the fixation point with an angle ranging from $-180°$ to $+180°$ (in steps of $15°$). We tested each network (VGG16 or RESNET101) either with raw images or with retinotopic mapping (Cartesian or Retinotopic). We observed a symmetry with respect to the horizontal flips and only show the positive angles here. This shows that VGG16 has a degraded performance compared to RESNET101, and notably that rotating images may have an adversarial effect on categorization performance, an effect which is less observed for RESNET101.

## 1.2 OUR PROPOSAL: INTRODUCING A BIOLOGICALLY INSPIRED MAPPING TO CNN INPUTS

The visual system in humans and many mammals is characterized by a substantial resolution disparity between the central area of the visual field (fovea) and the peripheral regions, where the number of photoreceptors decreases exponentially with eccentricity (Polyak, 1941). Such topographic maps, which transform the spatial relationships of sensory inputs, are a fundamental component of processing in species such as carnivores or primates (Kaas, 1997), suggesting that they confer evolutionary

advantages, although the precise computational benefits remain an active area of research. In primates (including humans), topographic maps are prominent in early visual processing areas of the brain. The retina projects to the lateral geniculate nucleus (LGN) in the thalamus, maintaining the spatial arrangement of photoreceptors. From there, a topographic map of the signals is sent to the primary visual cortex (V1), where adjacent neurons respond to adjacent locations in the retinotopic visual field. Higher visual areas such as V2 and V4 also exhibit this retinotopic organization.

A natural question is what advantages these non-isotropic visual inputs confer on information processing. Numerous hypotheses have been proposed regarding the role of this visual field deformation. One primary explanation is the coupling of foveal inputs with visual exploration: a retina with a fovea allows efficient visual processing if the eye can actively move and focus attention on specific points of interest. In addition, these precise mappings are thought to facilitate efficient parallel processing of spatial features. Topographic organization also facilitates connections within and between processing areas that respect the geometry of the sensory epithelium and minimize global wiring length. This supports functions such as the integration of local feature analysis into global perceptual representations. However, the function of this topography is still debated, with one possibility being that it is merely an artifact of the scaffolding that operates during development (Weinberg, 1997).

Here, we propose to take advantage of this biologically inspired preprocessing approach. Our goal is to build zoom and rotation invariance directly into the architecture of the network. Inspired by retinotopic mapping in visual pathways, we apply a transformation to input images such that pixel shifts in the transformed domain correspond to rotations or scale changes in the original spatial domain. By then using a convolutional network to process images in this transformed space, the local connectivity of the convolutions inherently imparts relative rotation and scale invariance to the model. Rather than expecting learned filters alone to discover such invariances post hoc, our approach maps rotations and zooms to translations at the input level. We hypothesize that this biological mapping front-end will improve the CNN's robustness to controlled geometric perturbations of images.

### 1.3 CONTRIBUTIONS OF THE PAPER

This paper makes several contributions:

- We show that popular off-the-shelf CNNs can be substantially perturbed by a simple rotation.
- We introduce a biologically inspired mapping to CNN inputs, inspired by retinotopic mapping in visual pathways, to build zoom and rotation invariance directly into the architecture of the network.
- We demonstrate the effectiveness of transfer learning models pre-trained on standard datasets to then classify inputs transformed in this way, without the need for retraining from scratch.
- We show that our approach induces significant robustness to controlled geometric perturbations compared to baseline models, confirming that the bank of convolutions in this transformed domain confers relative invariance as intended.
- We show improved object localization despite distortions, suggesting that retinotopic transformation may benefit tasks beyond classification
- We show that this biologically inspired mapping offers promising directions for future research exploring geometric deep learning, robust vision systems, and computational models of early visual processing.

## 2 INTRODUCING BIOLOGICALLY INSPIRED RETINOTOPIC MAPPING

Retinotopic mapping in humans results from the combined effect of the arrangement of photoreceptors in the retina and their output convergence via the optic nerve. This causes nearby regions of the visual field to activate adjacent neural structures as signals travel from the retina to the brain, while giving more resolution to the central field of view. Simple parameterized transformations computationally model this biologically inspired retinotopic mapping.

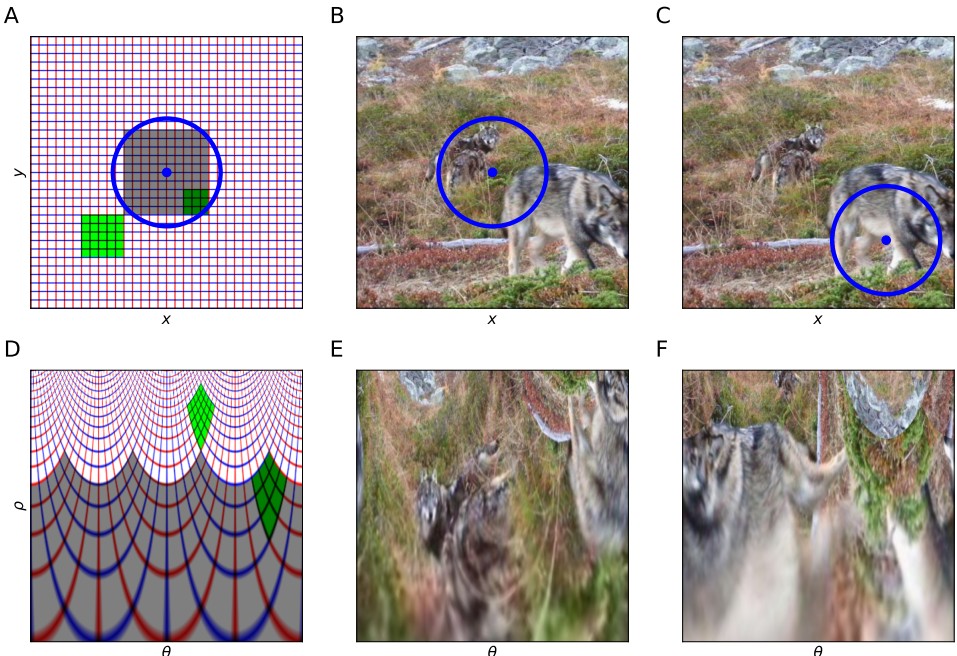

Figure 2: We illustrate the process of mapping input images defined in Cartesian coordinates to retinotopic space using a log-polar transformation with the fixation point (marked by a blue disk) and the approximate area of the fovea (blue circle). In **(A)**, the input image is defined as a regular grid representing the Cartesian coordinates $(x, y)$ by vertical (red) and horizontal (blue) lines. As shown in **(D)**, by applying the log-polar transform to this image, the coordinates of each pixel with respect to the fixation point are transformed based on its azimuth angle $\theta$ (abscissa) and the logarithm of its eccentricity $\rho = \log(\sqrt{x^2 + y^2})$ (ordinates). This transformation results in an overrepresentation of the central area and a deformation of the visual space. Note that the green square is translated in retinotopic space when it is scaled and rotated. When the transformation is applied to a natural image, as shown in **(B)**, there is a noticeable compression of information in the periphery, as shown in **(E)**. As shown in **(F)**, this representation is highly dependent on the fixation point, as indicated by the shift shown in **(C)** when the fixation point is moved to the right and down.

## 2.1 THE LOG-POLAR TRANSFORM AS A ROBUST RETINOTOPIC MAPPING

We implement the retinotopic mapping by transforming the Cartesian pixel coordinates into log polar coordinates. As shown in Figure 2, the location of each pixel in the log-polar domain $(\rho, \theta)$ is defined by

$$\rho = \log \sqrt{x^2 + y^2} \tag{3}$$

$$\theta = \arctan(\frac{y}{x}) \tag{4}$$

Where $(x, y)$ are the Cartesian coordinates. The increasing size of the receptive fields with eccentricity $\rho$ also parallels the decreasing retinal ganglion cell density in the periphery of the visual field. This retinotopic mapping has a unique and important property - zooms and rotations in Cartesian space become vertical and horizontal shifts, respectively, in log-polar space. This mapping transforms these two geometric transformations into two independent dimensions, since rotation only changes the azimuthal angle $\theta$, while zooming affects the logarithmic eccentricity $\rho$ on the elevation axis.

Crucially, by applying convolutions such as those used in CNNs after the log-polar transform, the model inherently acquires relative rotational and scale invariance without having to explicitly learn these transformations. Together, these properties allow our model to pool the translational invariance of convolutions while respecting the geometry of retinotopic mapping, which is computationally advantageous in biological vision systems. This type of transformation is commonly used in computer

vision, especially for template matching (Araujo & Dias, 1997; Sarvaiya et al., 2009; Maiello et al., 2020) or robotics (Traver Roig & Bernardino, 2010; Antonelli et al., 2015), but is largely underutilized in deep learning, especially in CNNs.

This transformation is performed with the PYTORCH library (Paszke et al., 2019) through the use of the grid_sample () function, as it applies a grid to the pixels of the image in Cartesian coordinates. This function has been implemented particularly efficiently for its use in spatial transformer networks (Jaderberg et al., 2016). Thus, with such a log-polar grid, each pixel in Cartesian space is assigned a new position in retinotopic space. We set the number of sampled angles ($N_\theta$) and the number of sampled eccentricities ($N_\rho$) to 256 to obtain an output image with a resolution of $256 \times 256$, which was also used during training. The resolution of the grid was set to the same dimension as the input images in the classical model to avoid bias in the evaluation due to image resolution. All $\theta$ values are within a linear distribution in $[0; 2\pi]$, while $\rho$ values are within a logarithmic distribution in $\log_2(r_{\min}; r_{\max})$. In practice, we use a log-polar grid with a radius of $r_{\max} = 0$ and a minimum radius of $r_{\min} = -5$. This choice of parameters allows us to focus on the central area of the image, which is the most informative part of the image. After analyzing various $r_{\min}$ parameters (performed with a central fixation point), we set $r_{\min}$ to $-5$; $r_{\max}$ fixes the radius and depends on the desired sub-sampling size. For example, setting $r_{\max}$ to 0 gives a maximum range of $\rho$ values within a $\log 2$ distribution in $[0.03; 1]$.

## 2.2 DATA SETS

We have selected two datasets for our study. The first dataset is that from the IMAGENET (Russakovsky et al., 2015) challenge, which is widely used due to its extensive collection of images and associated labels. This dataset offers rich semantic links, enabling the construction of task-specific datasets. However, it is worth mentioning that IMAGENET exhibits certain biases, particularly with objects being centered in many images. This characteristic makes it suitable for applying a log-polar transformation, where information is concentrated around the fixation point, which is considered the center of the image during training.

Despite its advantages, IMAGENET has limitations for localization tasks. For instance, it lacks multilabels, meaning there is only one label per image, and the proportion of bounding boxes relative to the image size is relatively small, which can limit the impact of certain analyses. To address these limitations, we also utilize the ANIMAL 10K (Yu et al., 2021) dataset. This dataset provides key points for each animal present in an image. By fitting Gaussians to these key points, we can generate heat maps centered around the label of interest, which, in this case, is 'animal' (see Figure 7 in the Appendix). This approach enables us to improve localization and better analyze the distribution of animals in the images.

## 2.3 RETRAINING, TRANSFER LEARNING AND EVALUATION OF RESULTS

For the first dataset, we used the standard pre-trained VGG16 and RESNET101 networks provided by the PYTORCH library. We explore two network configurations: one with a retinotopic mapping at the input and one without. To evaluate the efficiency of introducing the retinotopic mapping, both networks are retrained (or fine-tuned) with retinotopic inputs and compared to the legacy networks on the IMAGENET dataset. In a first generation, images were transformed by setting the fixation center to the center of the image, exploiting the fact that images in the IMAGENET dataset are *a priori* more likely to be centered. While giving good results, a number of visual objects were not centered, giving raise to some variability. As information degrades rapidly with eccentricity from the fixation point, the dataset was pre-processed by using the bounding boxes provided with the IMAGENET dataset to center the images. We used a fixation point defined as the center of the bounding box of the label of interest. These new images were used to train these networks in a second generation. This approach is more robust to the position of the visual object, but requires reliable bounding boxes. This procedure yielded more accurate results, and we used this procedure in the paper. It should be noted that even when using the full image, the networks were able to efficiently categorize the 1000 labels, demonstrating the specific robustness of the retinotopic referential, although the localization of these labels was not optimal.

Previously, it has been successfully demonstrated that using transfer learning to re-train networks VGG16 (Simonyan & Zisserman, 2015), allows its application to different tasks using the semantic

link that underlies IMAGENET's labels. It has also been shown that it is possible to predict the likelihood of a trained network on a newly defined task (such as categorising an animal) using this semantic link that connects the outputs of a pre-trained network to a library of labels, in particular to learn to categorize images containing or not an animal (Jérémie & Perrinet, 2023). Therefore, based on these findings, we used these networks to perform a categorization task on the second dataset, ANIMAL 10K (Yu et al., 2021), seeking animals in the image.

Since the network is asked to perform inference over 1000 labels during training, we implemented loss using the cross-entropy loss from the PYTORCH library. We used the stochastic gradient descent (SGD) optimizer from the PYTORCH library and validated parameters such as batch size, learning rate, and momentum by performing a sweep of these parameters for each network. During the sweep, we varied each of these parameters over a given range while leaving the others at their default values for 1 epoch on 10% of the entire IMAGENET training dataset. We chose the parameter values that gave the best average accuracy on the validation set: batch size = 80, learning rate = 0.001, momentum = 0.9.

## 3 EMPIRICAL RESULTS

### 3.1 TRAINING ON TRANSFORMED IMAGES

We fine-tuned the pre-trained VGG16 and RESNET101 networks on the IMAGENET dataset with retinotopic transformed images. We compared the performance of the networks trained on the original dataset (with a Cartesian mapping) with the performance of our retrained network on the transformed dataset defined in retinotopic space. We found that the networks retrained on the transformed images had similar categorization accuracy as the network pre-trained on regular images (see Figure 1-A & B). This is surprising given that the networks used in this retraining process were previously trained on regular images and that images with a log-polar transformation show a high degree of distortion, in particular a compression of visual information around the fixation point and a degradation of textures in the periphery, see Figure 2. One hypothesis is that the degradation of texture during the frame of reference change may cause the network to rely on shape rather than texture. However, it remains to be investigated whether there is a qualitative change in the features underlying the categorization.

### 3.2 ROBUSTNESS TO ROTATIONS

We then tested the robustness of the networks to rotation. We applied a rotation to the input image around the fixation point with an angle ranging from $-180°$ to $+180°$ in steps of $15°$. We tested each network (VGG16 or RESNET101) either with raw images or with retinotopic mapping (i.e., Cartesian or retinotopic). We first observed that VGG16 has a degraded overall performance compared to RESNET101 and that for both networks, image rotation has a detrimental effect on categorization performance, yet less pronounced for RESNET101. We also observed a symmetry of results corresponding to horizontal flip invariance in images, such that we only show results for half of rotation angles.

Noteworthy, these results show that while the VGG16 network retrained and tested on regular images showed some degradation for different rotations, the categorization results were much more invariant for the network including a retinotopic mapping (see Figure 1-C). This phenomenon is a consequence of the translation invariance imposed by the structure of the CNNs. Applied to retinotopic mapping, this translation invariance in retinotopic space is translated into rotation and zoom invariance in visual space (Araujo & Dias, 1997). The performance of RESNET101 with either Cartesian or retinotopic mapping is similar. Surprisingly, although this network was not designed a priori for retinotopic images, we observe a slight but consistent advantage for retinotopic mapping in this retrained network.

### 3.3 VISUAL OBJECT LOCALIZATION

The CNNs described above are designed to categorize images by providing a likelihood value for each of the 1000 labels. This likelihood is a probability (i.e., a scalar between 0 and 1) that predicts the probability that the label is present in the image. This allows us to make a binary decision

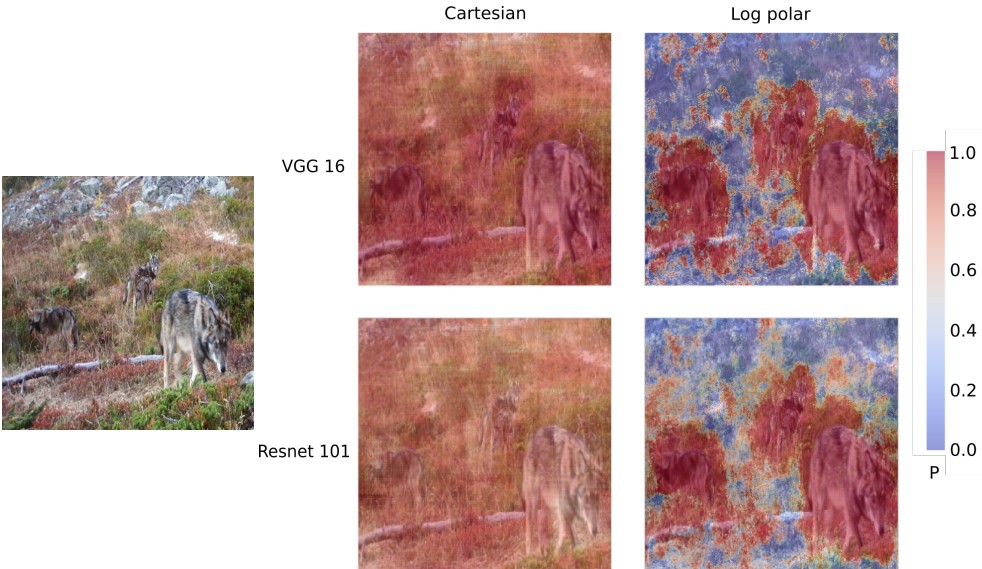

Figure 3: Likelihood maps computed on a representative image using $256 \times 256$ points of fixation, (**top**) with a VGG16, or (**bottom**) a RESNET101) network, trained and tested either on regular images (**left**) or on images mapped on the retinopic space using a log-polar grid (**right**). Red predicts the presence of the label 'wolf', blue its absence.

("present" or "not present"), e.g. by selecting the label corresponding to the highest likelihood. In our setting, we can also take different views of a large image and compute the likelihood for each of them, allowing us to compare which view provides the best likelihood ("bootstrapping"). Views can consist, for example, of focusing on regions of the image centered on different fixation points, with the fixation points aligned on a regular grid in visual space.

We used two parameters to define these maps: the first parameter is the resolution of the fixation point grid. The second is the size of the samples clipped at each of these positions, defined as the ratio of the log-polar grid radius of the input to the total input size or Cartesian grid size, since the grid is a square for Cartesian samples (for an illustration of this simple process, see Appendix Figure 6).

This sample is then transformed or not by the retinotopic mapping before being used as input for the corresponding network, see Appendix Figure 6-B & D. Conveniently, a collection of samples for different fixation points can be processed as a single batch. This protocol defines a likelihood map for any given network as the likelihood of categorizing the presence of a label of interest (here "an animal") inferred at regularly spaced fixation points in the image. A first qualitative exemple was process using $256 \times 256$ fixation points, and a 30% ratio, see Figure 3. As can be observed qualitatively in the example image, the Cartesian representation induces uniform activation across the whole image, while the retinotopic mapping induces differential activation inside and outside the object of interest.

To quantify this effect, we tested the networks with the likelihood map protocol on an $8 \times 8$ fixed grid of fixation points and an input sample with a 30% ratio see Table 1. Using the heat map extracted from the key points of the ANIMAL 10K (Yu et al., 2021) dataset as ground truth, "in" represents coordinates inside an animal (and "out" coordinates outside an animal, see Appendix Figure 7). For each point in the $8 \times 8$ grid, a likelihood value is obtained (probability of the presence of an animal). Next, we calculate the average likelihood for all points that are inside the zone corresponding to the animal (likelihood "in"), as well as the average likelihood for the zone that does not contain the animal (likelihood "out"). Next, we compare the values obtained in the "in" zone with those obtained in the "out" zone. A higher contrast indicates a better ability of the network to identify regions of interest in an image. For the VGG16, both performed well on the task, although the CARTESIAN tends to maintain a high likelihood outside the area of interest. For the RESNET101 networks,

Table 1: Likelihood maps results for the VGG16 and RESNET101 networks and as computed on the ANIMAL 10K (Yu et al., 2021) dataset. In: Average probability for a fixation point in a position where an animal is present. Out: Average probability for a fixation point outside the position where an animal is present. Ratio: Average probability in positions where there is an animal on the average probability outside positions where there is an animal (In/Out). We highlight for each network the mapping which reaches best performances for all three conditions.

|       | VGG16       |             | RESNET101   |             |
|-------|-------------|-------------|-------------|-------------|
|       | Cartesian   | Retinotopic | Cartesian   | Retinotopic |
| In    | **0.92**    | 0.76        | 0.73        | **0.80**    |
| Out   | 0.77        | **0.49**    | 0.69        | **0.54**    |
| Ratio | 1.19        | **1.59**    | 1.06        | **1.50**    |

the CARTESIAN version of the network seems to perform much less well than the RETINOTOPIC version in this exercise (see Table 1). If we consider a good categorization to be a high average probability on "in" coordinates (or a low probability on "out" coordinates), then in general networks using RETINOTOPIC meshes tend to be more contrasted than networks using CARTESIAN meshes, which is more apparent in the case of RESNET101, see Table 1).

Despite the greater diversity of poses, environments, and intra-class variation in ANIMAL 10K compared to IMAGENET, we found that they accurately highlighted regions containing the target animals. This suggests that our models learned general object-level features rather than dataset-specific cues.

## 3.4 TASK DEPENDENCE OF LOCALIZATION

We note that the likelihood maps generated by the model will vary depending on the label or class being predicted for a given input image. Specifically, the spatial distribution of high likelihood regions in the map differs depending on the label the model is trying to identify, or a class of objects defined by a set of labels. This suggests that the model learns to focus on different discriminative regions for different classes. The different likelihood maps provide insight into how the model's spatial attention changes according to the visual patterns that drive each classification decision. We tested this hypothesis on an example image by showing likelihood maps for the classes 'animal', 'dog', and 'cat'. As expected, the different regions are separated accordingly (see Figure 4).

The observation that likelihood maps vary by predicted class provides an exciting opportunity for future work. By modeling visual search as a path through these conditional spatial distributions, we may gain insight into how human gaze is affected during visual tasks. Attentional mechanisms have long been shown to guide eye movements in humans (Yarbus, 1961). This work therefore presents

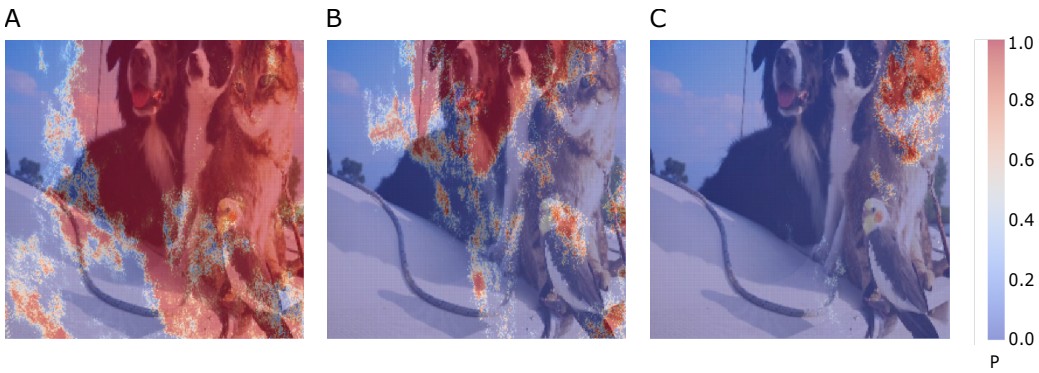

Figure 4: Likelihood maps generated by VGG16 networks trained and tested on retinotopic space using the log-polar grid. Searching for the (A) "animal", (B) "dog", or (C) "cat" label in an image.

promising opportunities to use deep networks as a means to unravel cognitive processes through computational modeling.

# 4 DISCUSSION

## 4.1 RETINA-INSPIRED MAPPING ENHANCES THE ROBUSTNESS OF CNNs

A first and main result of this study is to demonstrate the excellent ability of standard deep CNNs to deal with retinotopic inputs, even though this mapping represents a profound transformation of the visual inputs. The RESNET101 and VGG16 networks adapt easily to inputs where a large part of the image is highly compressed in the periphery and the spatial arrangement is significantly disturbed. The accuracy rates achieved with retinotopic inputs are equivalent to those of the original models.

In addition, the log-polar transformation has the advantage of better invariance to zoom and rotation. However, this invariance comes at the cost of reduced invariance to translation. For images that are not centered on the region of interest, one would have to shift the fixation point to the region of interest, similar to eye saccades.

The integration of a retinotopic mapping approach holds great promise for improving the efficiency and accuracy of image processing tasks. Our results are consistent with physiological data on ultra-fast image categorization, which show that human accuracy in recognizing briefly flashed images of animals is robust to rotation (Fabre-Thorpe, 2011). The log-polar compression used in our approach allows a seamless extension to larger images without a significant increase in computational cost.

## 4.2 RELATION TO PRE-ATTENTIVE MECHANISMS

As a second result, the definition of likelihood maps based on scanning the visual scene at a limited number of fixation points allows us to gain insight into the specificities of retinotopic processing: this transformation provides a more focused view, thus better separating the different elements of the image when focusing on its specific parts. As a result, it provides a proxy for the measurement of saliency with respect to the respective labels in the image.

In our case, it seems that the retinotopic mapping allows for a more precise localization of the category of interest, e.g., an animal, compared to off-the-shelf pre-trained networks using a Cartesian representation. It also gives us insight into the features on which our networks actually rely. Such information can be compared with physiological data (Crouzet, 2011), used to design better CNNs, and ultimately allow physiological tests to be proposed to further explore the features needed to classify a label of interest. In particular, by focusing on the point of fixation with the highest probability in the likelihood maps, we could consider refining the training of the network to our retinotopic mapping.

## 4.3 INTRODUCING EYE MOVEMENTS

Building on these observations, simulating human saccadic eye movement patterns during visual tasks provides an exciting opportunity to gain further insight into these mechanisms using such networks. A protocol that iteratively classifies image patches corresponding to foveated regions in a manner that mimics eye movements could reveal how network performance is spatially modulated across the visual field. Comparing classification accuracy under different saccade strategies, such as selecting the most uncertain or most likely location at each step, would provide valuable information about how network attention operates. This framework also allows the modeling of popular biological saccade strategies, allowing direct comparison with human visual search behavior.

Overall, implementing foveated classification with algorithmic saccades would provide a powerful method for validating existing attentional mechanisms in these networks, as well as inspiring new architectural innovations through embodied, task-driven visual attention modeling. Finally, the implementation of this robust categorization, coupled with refined localization of a label of interest and optimal saccade selection, could allow us to extend this study to a more complex task. One such task is visual search (i.e., the simultaneous localization and detection of a visual target), and the likelihood maps could provide the underlying pre-attentive mechanisms on which its effectiveness seems to depend.

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

# A APPENDIX

## A.1 ROBUSTNESS TO ROTATIONS ON THE ANIMAL 10K DATASET

To evaluate the robustness of our models to geometric transformations such as rotations beyond the IMAGENET dataset (see Figure 1), we performed additional experiments on the ANIMAL 10K dataset. This dataset contains 10,000 images from a variety of animal species and presents a challenging classification task due to high intra-class variability in pose, size, orientation, and environment.

As with IMAGENET, we first obtained baseline classification accuracy on ANIMAL 10K images without perturbation. We then systematically rotated each test image through the same range of rotation angles. At each rotation value, we computed the average accuracy over the entire test set in order to characterize performance as a function of viewing angle.

The results on ANIMAL 10K broadly mirror those on IMAGENET, with standard CNNs showing a sharp decline in accuracy for even small rotations away from the trained orientation, particularly away from the cardinal angles (see Figure 5). However, models incorporating our retinotopic preprocessing approach maintain significantly higher performance over the full range of orientations. This demonstrates that the benefits of rotational robustness generalize beyond the original dataset and image characteristics.

Testing on ANIMAL 10K, with its greater pose diversity and difficulty, reinforces the conclusion that our method improves invariance to natural, ecologically valid geometric transformations beyond what standard CNNs alone can achieve.

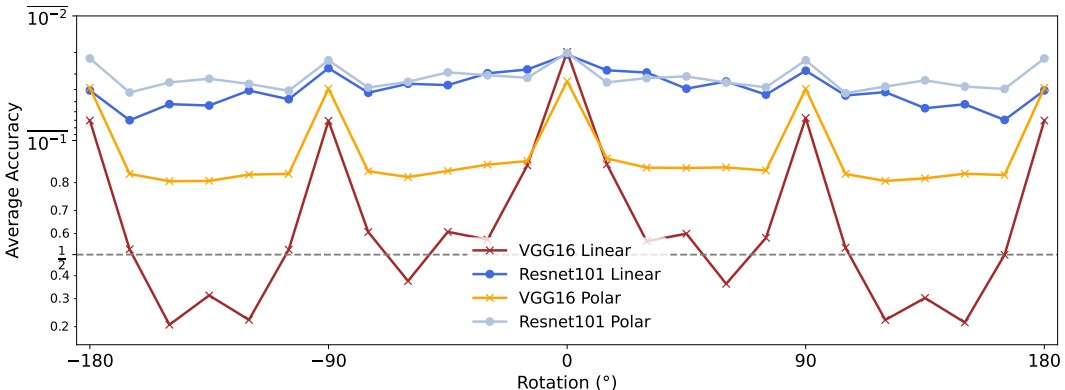

Figure 5: Average accuracy over the ANIMAL 10K (Yu et al., 2021) dataset, shown for both retrained and pre-trained networks with different input image rotations. The rotation is applied around the fixation point with an angle ranging from $-180°$ to $+180°$ (in steps of $15°$). We tested each network (VGG16 or RESNET101) either with raw images in Cartesian coordinates or with retinotopic mapping. The dotted line represents chance level.

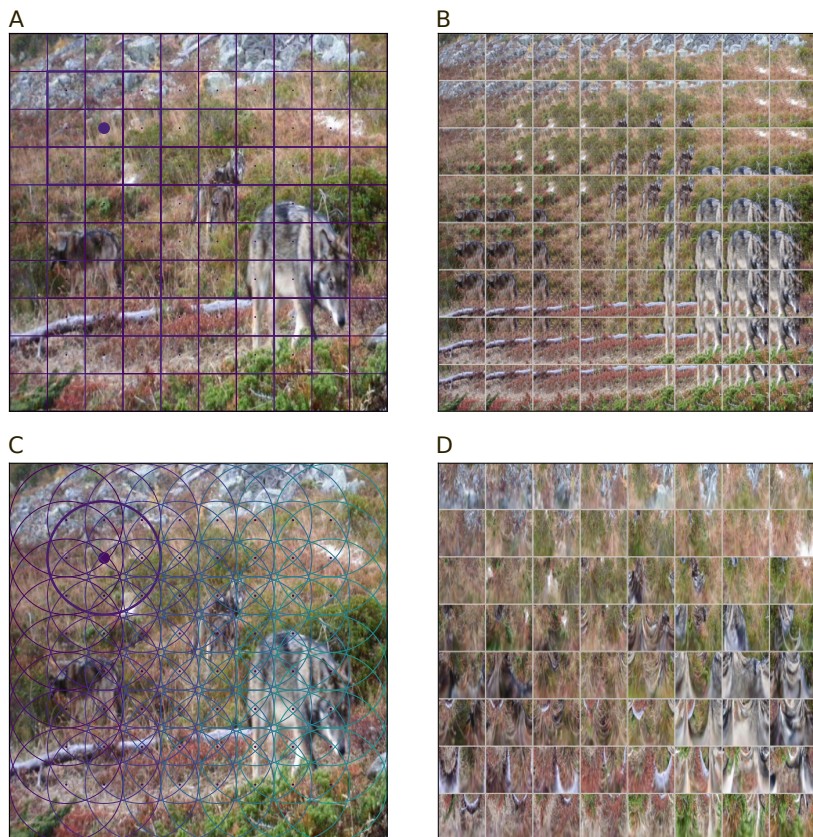

Figure 6: Generating different views of a single image to compute likelihood maps. **(A)** For the networks with Cartesian inputs, we used a regular grid of $8 \times 8$ fixation points, which allows samples to be cropped, highlighting a particular view. As shown in **(B)**, this produces a batch of images that can be used to generate likelihood maps. **(C)**) We used a similar grid to generate batches of retinotopic inputs, as shown in **(D)**). In **(B)** & **(D)** each sample corresponds to $30\%$ of the input.

## A.2 CREATING MULTIPLE VIEWS FOR LIKELIHOOD MAPS

To create likelihood maps to test our trained models, we generated batches of transformed input views for each full-scale test image. Specifically, we created a grid of fixation points as a grid of linearly spaced points on the horizontal and vertical axes.

We then cropped a sample of the image centered on each fixation point, with the size of the sample defined by the ratio of the log-polar grid radius to the total input size (or the Cartesian grid size, since the grid is a square for Cartesian samples). We then transformed each sample with or without the retinotopic mapping and ran the batch of transformed images through the model to obtain a likelihood map for that image-label pair(see Figure 7). We repeated this process for each image-label pair in the test set(see Figure 8).

## A.3 GROUND TRUTH HEAT MAPS

To quantify how accurately our models are able to localize objects within images, we generated ground truth heatmap labels for the datasets. For each image from ANIMAL 10K containing a set of keypoints, we created a Gaussian heatmap centered on those points, with the peak value set to 1 and values falling off with a standard deviation proportional to object size. This results in heatmap labels between 0 and 1 that capture the true spatial extent and location of the target object within each image (see Figure 7).

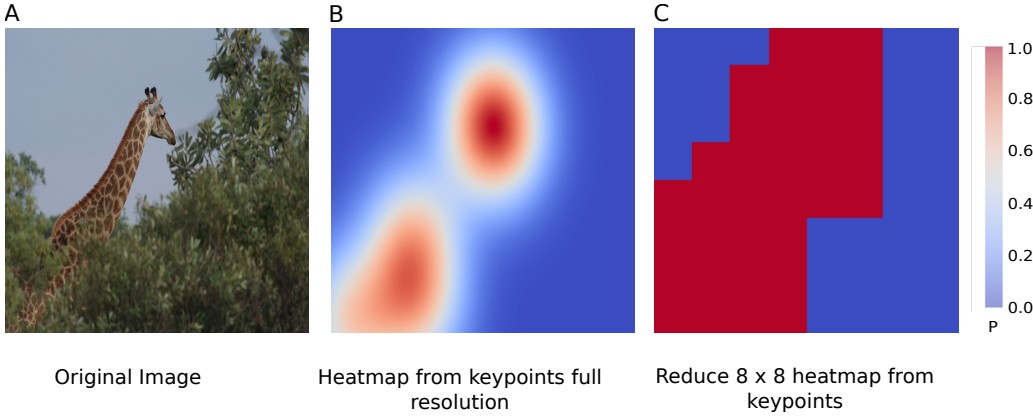

Figure 7: (A) The original image of the ANIMAL 10K dataset. (B) A heat map constructed by fitting Gaussians to the key points of the ANIMAL 10K dataset. (C) The heat map constructed in (B) is normalized and reduced to a resolution of $8 \times 8$ to be used as ground truth when evaluating the heat map. A threshold (0.2) is applied to reduce the heat map field to the assumed contour of the animal.

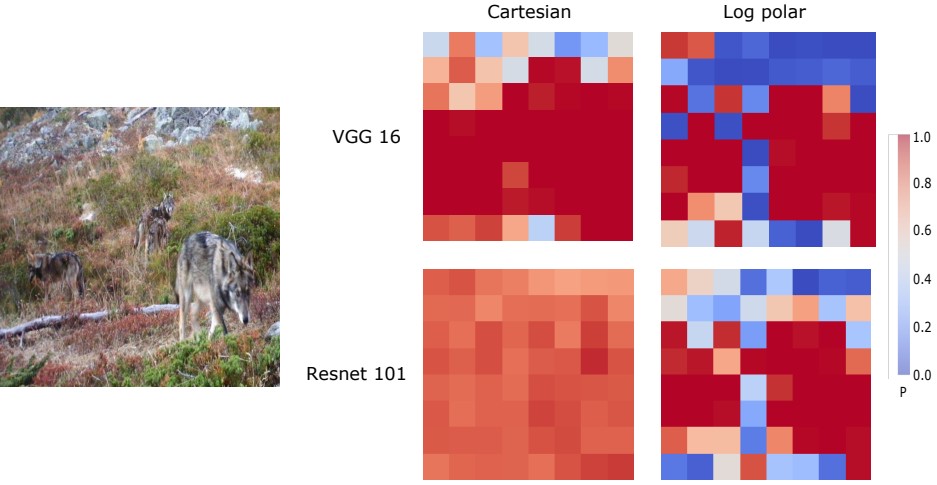

Figure 8: Likelihood maps computed on a representative image using $8 \times 8$ points of fixation, (as computed on the whole ANIMAL 10K (Yu et al., 2021) dataset), (**top**) with a VGG16, or (**bottom**) a RESNET101) network, trained and tested on either regular images (**left**) or images mapped in retinopic space using a log-polar grid (**right**). Red predicts the presence of the label 'animal', blue its absence.

