# OpenReview forum: "Enhancing Robustness of Visual Object Localization by Introducing Retina-Inspired Mapping to Convolutional Neural Networks"
_ICLR.cc/2024/Conference — ICLR 2024 Conference Withdrawn Submission_

### Official Review · Reviewer_Jd7k · 2023-10-20

**Soundness:** 2 fair
**Presentation:** 3 good
**Contribution:** 1 poor
**Rating:** 1
**Confidence:** 4

**Summary:**

The authors use a log-polar representation of images to train rotation- and scale-invariant image classifiers. They find that the log-polar versions of VGG-16 and ResNet-101 are more robust to image rotations than their (standard) Cartesian versions. They also find that placing the center of the log-polar representation at different places in the image can be used for object localization.

**Strengths:**

+ More human-like vision systems are an interesting and potentially important research direction to improve robustness
 + Well-written, straightforward to follow

**Weaknesses:**

### 1. Lack of novelty

The authors' approach is basically a Polar Transformer Network (Esteves et al., ICLR 2018). Unlike the original work (which they don't cite) they don't even address the question of how to determine the polar original, but just take labels from the ImageNet dataset.


### 2. Robustness is trivial

The authors motivate their work by adversarial robustness, but the actual approach doesn't get there at all. Instead, they test robustness against rotation, which is trivially better in a Polar Transformer Network because it is equivariant to rotation and scale by construction. Thus, I don't see how this approach brings us any closer to robustness in deep networks.

**Questions:**

None.

---

> ### Author Response · Authors · 2023-11-16
> **difference of our work with the Polar Transformer Network**
>
> Thanks for pointing toward the Polar Transformer Network (Esteves et al., ICLR 2018) which has indeed a clear relation with our work in the way the image is transformed.
>
> However, we would like to point out that:
> - we are not interested in guessing the focal point as in the PTN, our main scientific question is quite different, and tries to uncover why a retinotopic transform would be useful in biology,
> - our domain of application is quite different from small grayscale digits and we use full-scale, cor images from ImageNet or Animal10k
> - our results show different emerging properties, and notably a better localization of objects.

---

> > ### Comment · Reviewer_Jd7k · 2023-11-16
> >
> > Isn't the localization trivial, because you're essentially zooming into the region that is being focused because of the log-polar transform? If you don't have a mechanism that decides where to focus, how is the improved localization helpful? Any object detection/instance segmentation system will localize orders of magnitude better.

---

> > > ### Author Response · Authors · 2023-11-22
> > > **many thanks for your time**
> > >
> > > The paper aimed at responding at these questions. I would be glad to see any other system where "Any object detection/instance segmentation system will localize orders of magnitude better."
> > > Any?

---

### Official Review · Reviewer_XnPR · 2023-10-31

**Soundness:** 2 fair
**Presentation:** 3 good
**Contribution:** 2 fair
**Rating:** 3
**Confidence:** 4

**Summary:**

The authors investigate the potential of foveated vision—a feature in biological systems—in improving the performance of off-the-shelf CNNs. This is achieved by applying a log-polar transformation to the input images and retraining the CNNs. While the model trained with retinotopic inputs demonstrated similar classification performance compared to its standard counterpart trained with Cartesian images, it showcased superior robustness to image zooms and rotations, as well as improved classification localization.

**Strengths:**

- The writing is well-structured and easy to follow.
- The method is clean and solid.
- The results on visual object recognition are inspiring and may suggest the proposed approach can be applied to a dynamic visual model that incorporates the human saccadic eye moments.

**Weaknesses:**

- The novelty of this work is unclear to me, especially given the prior existence of similar concepts, such as Polar CNNs [1]. There are also many prior works applying foveated images to deep neural networks, and evaluated their robustness to adversarial attacks [2,3]. These related works are not discussed in the paper.
- The robustness on rotated images is expected given that the CNNs trained with polar-transform images inherently extract rotation-invariant features. Besides, this idea was also already validated in [1].
- The visual object localization experiments, while insightful, are not surprising and may result from an "unfair" comparison. Given the non-uniform sampling of the retinotopic input, it naturally narrows the "effective" FOV when you use the exact same 8x8 grid to compare Cartesian input vs. Retinotopic input.

**Questions:**

- Why rotational invariance is considered as a biologically plausible property? Previous studies [4] suggest that visual recognition in humans depends on the viewing angle. Also, the retinotopic mapping in the human visual system does not lead to an inherent invariance to image rotations.
- When retraining models such as VGG116 and ResNet101 with log-polar transformed input, did you apply circular padding for the $\theta$ axis?
- Did the kernels in CNNs retrained by log-polar transformed images also manifest meaningful feature extractors (e.g., edge detection in lower layers and shape recognition in higher layers)?

minor points: It's unconventional (and incorrect in my opinion) to refer to rotated images as an "attack".

**Referebce**:

1. Esteves, C., Allen-Blanchette, C., Zhou, X., & Daniilidis, K. (2017). Polar transformer networks. ICLR 2018.
2. Luo, Y., Boix, X., Roig, G., Poggio, T., & Zhao, Q. (2015). Foveation-based mechanisms alleviate adversarial examples. ICLR 2016.
3. Vuyyuru, M. R., Banburski, A., Pant, N., & Poggio, T. (2020). Biologically inspired mechanisms for adversarial robustness. Advances in Neural Information Processing Systems, 33, 2135-2146.
4. Lawson, R. (1999). Achieving visual object constancy across plane rotation and depth rotation. Acta psychologica, 102(2-3), 221-245.

---

> ### Author Response · Authors · 2023-11-16
> **Thanks for the review**
>
> Many thanks for the time taken to review our paper and the fair evaluation. By politeness, let us answer some of your questions:
>
> * indeed we had not discussed these works as our objective was slightly different: we are trying to assess *why* retinotopy may be useful in some species (and lacking in others, like rabbits). in the context of ICLR, we admit we should have included those papers.
>
> * concerning localization, it is not surprising indeed (it's the believed function of foveation after all), yet our results provide a (first?) quantitative evaluation for this "unsurprising" claim - and in particular in comparison to the output to linear inputs
>
> * indeed, we have padding in the polar axis - it does not change much to the performance (2%) yet it is mathematically sounder.
>
> * "Did the kernels in CNNs retrained by log-polar transformed images also manifest meaningful feature extractors (e.g., edge detection in lower layers and shape recognition in higher layers)?" This is an excellent point and sums up very well our objective : we expect features to be more explainable and categorization to be less dependent on textures. More experiments are necessary to prove that points.
>
> * Considering "It's unconventional (and incorrect in my opinion) to refer to rotated images as an "attack"." - we agree that that the drop of accuracy is surprisingly large for these off-the-shelf CNNs that are used in widespread applications  (I find no reference for that finding), yet this is an attack: knowing a model, optimizing for a parameter to achieve the least performance.

---

> > ### Comment · Reviewer_XnPR · 2023-11-18
> >
> > Thank you for your reply.
> >
> > First, I think that even if the purpose of this paper is to "assess why retinotopy may be useful in some species and lacking in others" (which was not really discussed), it is clearly inappropriate to ignore all previous literature on polar/retinal transformation models.
> >
> > Secondly, I still think my question was not answered. The localization effect may simply arise from an unfair comparison, because if the effective FOV in the two conditions are different the result would be trivial (as also raised by the other reviewer). As I mentioned in the first-round review, incorporating this with eye movements will be an interesting direction in pursuit of a more bio-plausible model, but unfortunately the authors didn't experiment with this.
> >
> > Therefore, while the topic is interesting, I don't think the authors provide enough evidence to support that their work is a novel contribution to the field, and I decide to lower my score.

---

> > > ### Author Response · Authors · 2023-11-18
> > > **clarification**
> > >
> > > First, I fully agree that it is clearly inappropriate to ignore all previous literature on polar/retinal transformation models, and that the one on which we have focused was not that that you suggested, I am sorry for that. Note that the literature that was suggested by the other reviewers is different from one reviewer to the other (except (Esteves et al., ICLR 2018) which is irrelevant for us as it is a STN which infers the best fixation point and that is limited to small images).
> > >
> > > Second, the eye movements were not the scope of this paper (given a 8 page limit) and the comparison between Cartesian and Retinotopic was made fair by comparing different metrics inside and outside the object. See paper.
> > >
> > > I do not see statistically the need of lowering the score from 5 to 3, given the scores from other reviewers, but respect your "choice". Many thanks for your time.

---

### Official Review · Reviewer_GpAi · 2023-11-02

**Soundness:** 3 good
**Presentation:** 2 fair
**Contribution:** 2 fair
**Rating:** 3
**Confidence:** 5

**Summary:**

This paper introduces the idea that having a log-polar transformation at the start of a CNN (as a pre-processing stage) improves the general robustness/accuracy of an animal localization task, while maintaining classification performance. The Authors also are interested in shedding light on the question of what are the computational advantages (if any) of foveation.

**Strengths:**

* This paper's main strengths are the scientific questions authors are trying to address. However, I am not sure that the logic is fully correct and the experiment prove their main point. The question however is very interesting, and not addressed enough in the field: Why do humans foveate, and what advantages can machines get of such spatially-adaptive computation? This it the type of questions that are very hot in the emergent field of NeuroAI.
* The authors use other datasets other then ImageNet (they use Animal-10k) -- though see Weaknesses, I am not sure if this is a good decision.
* The notion of adding scale and rotational invariance as a pre-processing transform is interesting, but I also think that this has been addresses in other ways with multi-scale transformer archtiectures such as CrossViT (Chen et al. 2021)

**Weaknesses:**

* Not enough experiments. Also not sure how the brittleness of lack of rotational invariance (Figure 1) is later addressed in Table 1. Unless Figure 1 already shows the final result with the color-tone but after re-reading the caption, I don't think this is the case.
* Weird experimental selection : Why Animal-10k dataset over something like ImageNet (Objects) or Places (Scenes)?

--------

There are a lot of **relevant missing papers** regarding the question of "What is the purpose of Foveation?" and similar in the previous work section (and that can contribute to the discussion). See below:

Key Missing Critical References:
- Deza & Konkle. ArXiv, 2021. Emergent Properties of Foveated Perceptual Systems.
- Wang & Cottrell. Journal of Vision, 2017. Central and peripheral vision for scene recognition: A neurocomputational modeling exploration.
- Cheung, Weiss & Olshausen. ICLR 2017. Emergence of foveal image sampling from learning to attend in visual scenes

Secondary, but also important References:
- Gant, Banburski & Deza. SVRHM, 2022. Evaluating the adversarial robustness of a foveated texture transform module in a CNN.
- Reddy, Banburski, Pant & Poggio. NeurIPS 2020. Biologically inspired mechanisms for adversarial robustness
- Wang, Mayo, Deza, Barbu & Conwell. SVRHM, 2021. On the use of Cortical Magnification and Saccades as Biological Proxies for Data Augmentation
- Harrington & Deza. ICLR, 2022. Finding Biological Plausibility for Adversarially Robust Features via Metameric Tasks

**Questions:**

- Table 1: Shouldn't the probably of fixation in animal and out animal sum to 1 in the aggregate? Or not necessarily?
- Figure 5 : What is accuracy here? Is it a percentage or a ratio? The top value of $10^{-2}$ would mean that the highest accuracy is 1% (0.01), or is this a typo? Should it be $10^2$ instead?
- Figure 6 (Supplement) looks strange. Why is the log-polar transform being computed locally per each small region, vs over the whole image given a point of fixation.
- Given the previous question. What is the point of Figure 6 given Figure 2 -- which seems like what the Authors are doing.

I am open to changing my score, perhaps I did not understand the authors key contributions, and they are welcome to address many of my concerns in their rebuttal.

I am not rejecting the paper due to lack of innovation (novelty) [The paper poses a really interesting question, and approach], but rather because I am not fully convinced or understand what the authors intend to show in the paper through their limited experiments -- including those in the Supplementary Material.

---

> ### Author Response · Authors · 2023-11-16
> **thanks for the review**
>
> By politeness, let us answer to some questions:
>
> * First, many thanks for your detailed review and the very useful pointers. We were missing some of them and did not cite the Wang & Cottrell (2017) one. Sorry for that omissions. There are other suggestions by other reviewers that are very relevant and we are trying to gather all that bibliography in a common shared bibliographic repository.
>
> * the  Animal-10k  dataset is relevant to us as it is more common to ask a human "is there an animal in the image?" rather than "is there a XXX in the image?" where XXX is "vending machine", "goldfinch", "axolotl" or "goblet" (taking some at random). our interest is more towards neuroAI, that is how can we better understand cognition using AI rather than reaching another SOTA.
>
> * "Table 1: Shouldn't the probably of fixation in animal and out animal sum to 1 in the aggregate? Or not necessarily?" these are likelihood given by the model for any position
>
> * "Figure 5 : What is accuracy here? Is it a percentage or a ratio? The top value of  would mean that the highest accuracy is 1% (0.01), or is this a typo? Should it be  instead?" we used the notation to show the complement - the highest accuracy is 99%. we would have clarified this in the caption
>
> * "Figure 6 (Supplement) looks strange. Why is the log-polar transform being computed locally per each small region, vs over the whole image given a point of fixation." this is to show how the input image changes for different fixations points.
>
> * "Given the previous question. What is the point of Figure 6 given Figure 2 -- which seems like what the Authors are doing."  Figure 2 shows one change of fixation point, figure 6 a grid of those.

---

### Official Review · Reviewer_EiMB · 2023-11-02

**Soundness:** 2 fair
**Presentation:** 3 good
**Contribution:** 2 fair
**Rating:** 3
**Confidence:** 5

**Summary:**

The authors propose the incorporation of foveated visual processing into deep convolutional neural networks (CNNs). The authors pre-process images with a log-poral mapping before passing the images through off-the-shelf CNNs for (re)-training and evaluation. The authors show that the incorpoated foveated processing improves the robustness to scale and rotation perturbations while retaining classification accuracy on non-perturbed inputs. The authors also show that the foveated network produces improved classification localization when the fovea center was moved, which is not possible to perform in standard non-foveated CNNs.

**Strengths:**

+ This submission proposes a simple extension of pretrained off-the-shelf CNNs with a log-polar retinotopic transform which enhances the rotation and scale invariance of the learned representations while retaining accuracy on non-perturbed upright images.
+ There are interesting discussion sections on connecting the proposed architecture with pre-attentive mechanisms and eye movements in visual processing.
+ The writing in this paper is very clear and the visualizations (esp. Fig 2 illustrating log-polar transforms) are helpful to improve the readability of the submission. In my opinion, the paper is quite easily accessible for both computer vision and neuroscience audience.

**Weaknesses:**

- The proposed approach lacks novelty; contrary to the authors claiming to introduce the biologically-inspired log-polar retinotopic mapping to CNN inputs, this has been explored in the past [1, 2, 3], the authors haven't added this related work in their paper and state that log-polar transforms in CNNs are largely underutilized. [2] explores object localization performance on rotated images which is one of the core premises of the current submission.
- The proposed work uses VGG-16 and ResNet-101 networks which are far behind the current state of the art in computer vision. This reduces the impact of the proposed work for machine learning. I would suggest the authors to please use more recent architectures with higher classification performance (on both clean images and rotated images) if they would like to make a strong contribution towards rotation invariant neural networks.
- While using off-the-shelf networks to show improved rotation and scale invariance, the authors are restricted from reporting how significant the observed gains are over multiple random seeds. I would suggest adding both stronger baselines (in relation to my previous point) and evaluating performance over multiple random initializations in order to provide a fuller picture of how important log-polar mapping is to rotation invariance. If the authors are to make this submission more exciting to the computer vision community, they must clearly state how such input transformations are helpful (and feasible) in a time where pre-training full models is less viable as the industry moves towards finetuning task-specific decoders on top of strong frozen backbones (which seem to possess strong generalization abilities already).
- Overall, I find this submission to not be making very exciting contributions to either computer vision or neuroscience communities and that it could be significantly improved before publication at ICLR.

References:
1. Remmelzwaal, L. A., Mishra, A. K., & Ellis, G. F. (2020, January). Human eye inspired log-polar pre-processing for neural networks. In 2020 International SAUPEC/RobMech/PRASA Conference (pp. 1-6). IEEE.
2. Cao, J., Bao, C., Hao, Q., Cheng, Y., & Chen, C. (2021). LPNet: Retina inspired neural network for object detection and recognition. Electronics, 10(22), 2883.
3. Gahl, M., Kulkarni, S., Pathak, N., Russell, A., & Cottrell, G. W. (2022). Visual Expertise and the Log-Polar Transform Explain Image Inversion Effects.

**Questions:**

Please refer to my weaknesses section above.

---

> ### Author Response · Authors · 2023-11-16
> **thanks for the pointers**
>
> * many thanks for providing with related bibliography which we were not aware of. The Polar Transformer Network (Esteves et al., ICLR 2018) is another relevant publication.
>
> * we have used VGG-16 and ResNet-101 as 1/ they are generic enough to provide a fair result when comparing with or without retinotopic mapping, 2/ they use less energy, and the cost of transferring to bigger architectures (ViT etc) would not provide any qualitative wisdom *a priori*, 3/ we do not have access to clusters of GPU, rather off-the-shelf and prefer cross-validation to SOTA.
>
> * thanks for the suggestions on adding stronger baselines, this is something which we will add in the near future
>
> * we find it exciting and hope that a new version will convince you in the near future ;-)